# Rethinking HIV care for youth: Insights from qualitative research with youth in Chad

Esias Bedingar [1,2]*, Ferdinan Paningar[3], Ngarossorang Bedingar[4], Eric Mbaidoum[5], Naortangar Ngaradoum[5], Rifat Atun[1‡], Aisha K. Yousafzai[1‡]

1 Global Health and Population Department, Harvard T.H. Chan School of Public Health, Boston, Massachusetts, United States of America, 2 Alma, Centre de Recherche en Systèmes de Santé, N'Djamena, Chad, 3 Bucofore, N'Djamena, Chad, 4 Croix Bleue Tchadienne, N'Djamena, Chad, 5 Réseau National des Personnes Vivants avec le VIH, N'Djamena, Chad

---These authors contributed equally to this work.
‡ RA and AKY also contributed equally to this work.
* esias.bedingar@gmail.com

## Abstract

Youth ages 15–24 years are significantly impacted by the HIV/AIDS epidemic, representing approximately 37% of new infections globally. This demographic is especially vulnerable in sub-Saharan Africa, where over 80% of HIV-positive youth reside. In Chad, youth face barriers to effective HIV care, including high prevalence rates, particularly among young women, and substantial disparities across regions. Despite overall reductions in new HIV infections, youth remain disproportionately affected, necessitating targeted interventions to improve HIV care outcomes. This study represents a secondary data analysis derived from a parent study that employed a grounded theory design to develop theory inductively. The secondary analysis aimed to generate an in-depth understanding of the pathways to care for youth in Chad, exploring barriers and facilitators across the care continuum, from diagnosis to antiretroviral therapy (ART) adherence. Data were collected through focus group discussions with 52 youth and 48 service providers, including healthcare workers and community actors. Data were transcribed, translated, and analyzed assisted with ATLAS.ti software (Version 7.6.3). Youth identified barriers such as financial constraints, logistical challenges, and fear of stigma, while facilitators included peer support and specialized HIV care facilities. Healthcare workers emphasized the critical role of psychosocial counseling and confidentiality in facilitating youth engagement with HIV services. They also highlighted critical challenges, including the lack of accessible and youth-friendly services, and the need for continuous education to reduce stigma. The findings underscore the importance of tailored, youth-friendly interventions that address these challenges, foster supportive environments, and integrate youth and provider perspectives. We recommend redesigning healthcare services to improve accessibility, reduce stigma, and provide continuous psychosocial support, ultimately enhancing the HIV care continuum for youth in Chad and similar contexts.

**Data availability statement:** The data underlying this article cannot be shared publicly due to ethical restrictions imposed by the Institutional Review Board (IRB) at the Harvard T.H. Chan School of Public Health (Protocol #: IRB23-1743). During the ethical approval process, it was specified that all data would remain confidential, and access to recordings and transcripts would be restricted to the research team to protect participant privacy and sensitive information. As part of the informed consent process, participants were assured both verbally and in writing that their contributions would be kept confidential. While it was mentioned that non-identifiable quotes may be used to report findings, participants did not consent to making their raw data publicly available in any repository. Sharing de-identified data publicly would constitute a breach of the ethical assurances provided to participants during the study. Researchers who wish to access the data for verification purposes or secondary analysis may submit a formal request to the Harvard T.H. Chan School of Public Health IRB. Requests can be directed to the Office of Regulatory Affairs and Research Compliance at irb@hsph.harvard.edu, referencing Protocol #: IRB23-1743. Such requests will be evaluated to ensure compliance with ethical standards and the confidentiality agreements established with the study participants.

**Funding:** The study received funding from the Tessa Jowell Fellowship for Doctoral Research and Chad's Prime Minister's Office.

**Competing interests:** The authors declare that they have no known competing financial interests or personal relationships that could have appeared to influence the work reported in this paper.

**Abbreviavtions:** ART, antiretroviral therapy; AIDS, acquired immunodeficiency syndrome; CNLS, National AIDS Control Council; DHS, demographic and health surveys; FGD, focus group discussion; HIV, human immunodeficiency virus; PAR, participatory action research; RNTAP+, National Executive Secretary of the Chadian National Network of Associations of People Living with HIV; SRH, sexual and reproductive health; SSA, Sub-Saharan Africa; UN, United Nations; UNAIDS, Joint United Nations Programme on HIV/AIDS; UNICEF, United Nations International Children's Emergency Fund; WHO, World Health Organization; YLHIV, youth living with HIV.

## Introduction

Youth (ages 15–24) are a crucial demographic in the global fight against HIV/AIDS, comprising about 37% of new HIV infections worldwide [1–3]. Youth (10–24 years) are especially vulnerable, experiencing high HIV prevalence and suboptimal treatment outcomes [4,5]. This issue is pressing in sub-Saharan Africa, where over 80% of youth living with HIV (YLHIV) reside [6–8], with high rates of HIV-related mortality among young people [9]. In Chad, the HIV prevalence was 1.0% in 2022, down from 1.6% in 2010 and 1.4% in 2015, but varies significantly across provinces [8,10].

In Chad, young women face higher HIV infection rates compared to young men due to the "triple threat" of new HIV infections, sexual and gender-based violence, and early pregnancies [11]. In 2022, 26.3% of new HIV infections in Chad were among youth aged 15–24 [8]. Despite a 48% overall decrease in new HIV infections since 2010, youth remain disproportionately affected [8], highlighting a gap in prevention and care services tailored to this group [12].

Youth face numerous barriers in the HIV care continuum, from testing to achieving viral suppression, including stigma, transportation issues, and the transition from pediatric to adult care [13,14]. In addition to other barriers, young people in Chad face legal and policy-related obstacles to accessing HIV services. National regulations require individuals to be at least 18 years old to consent independently to HIV testing and treatment. Those under 18 must obtain parental or guardian consent, creating a significant barrier to confidential care. These challenges lead to lower linkage to care, higher attrition, and poor adherence to ART among youth compared to other age groups [15,16]. Limited studies indicate poorer outcomes for HIV-infected youth in the care pathway [17], with significantly lower retention rates in care before initiating ART and low adherence even after starting treatment [18].

Service delivery interventions such as counseling, peer support, financial incentives, and youth-friendly clinics show promise in improving linkage, retention, and adherence among youth [17,19,20]. Integrating biomedical prevention with behavioral, psychosocial, and structural approaches can enhance the HIV care continuum, especially in the context of Chad's complex sociocultural landscape.

This study aims to explore the experiences and perspectives of youth regarding their pathways to care, focusing on their journey from HIV diagnosis to ART initiation, retention, and adherence. By understanding the challenges and facilitators identified by participants, we seek to uncover strategies to enhance the continuum of HIV care for this vulnerable population.

## Methods

### Study design

This study is a secondary data analysis from a parent study employing grounded theory design [21,22]. The stages and processes employed for the parent study are shown in Fig A in S1 File. The original data were collected to understand youth's sensemaking and service utilization in the context of SRH and HIV care in Chad as presented in Fig B in S1 File.

## Sampling

Participants were selected using purposeful criterion sampling to ensure a diverse and relevant sample that represented a wide range of perspectives and experiences. Diversity was defined in terms of gender, HIV status (youth living with HIV and those without), and roles within the healthcare system (youth and healthcare providers). Relevance was determined by the participants' direct experience with HIV care and services, either as recipients (youth) or as facilitators/providers (healthcare workers and community actors). The sampling frame was bound to the province of N'Djamena to maintain contextual relevance while allowing for a variety of perspectives. Youth, aged 15–24 years, were recruited and divided into two groups: those living with HIV and those without HIV. The final sample included 52 youth, with an almost equal distribution of gender and HIV status. For youth living with HIV, we collaborated with the National Executive Secretary of the Chadian National Network of Associations of People Living with HIV (RNTAP⁺) who helped identify those willing and eligible for the study. Eligibility was defined as either a male or female ages 15–24 years living with HIV. A specific number of participants was planned to ensure manageable and meaningful group discussions. Random selection was conducted to choose participants from the pre-established list of eligible individuals provided by RNTAP⁺. A random number generator was used to ensure fairness and minimize selection bias while adhering to the planned sample size. This approach facilitated the selection of a representative and logistically feasible sample within the study's scope.

The study also included 48 service providers, divided into three categories, including nurses, doctors, and community actors to provide comprehensive insights into the healthcare environment. Each group of service providers was subdivided into three groups of four people for healthcare workers, and four groups of six people for community actors. For both nurses and doctors, the groups reflected health facility type, including public and specialized facilities for SRH and HIV care. Eligibility criteria for health facilities included the existence of pediatric, maternity and SRH service areas, as well as being frequently visited by youth. After selecting these facilities, the second step consisted of obtaining a list of nurses and doctors who had been working at the facility for at least six months to ensure sufficient exposure to youth. For community actors, the four groups included those existing in Chad such as peer educators, mentor moms, expert patients, and psycho-social counselors [23].

## Data collection

Data were collected through focus group discussions (FGDs) conducted from 24/03/2024 to 27/03/2024. The FGDs were designed to elicit detailed accounts of participants' experiences, beliefs, and behaviors related to HIV care and ART. Separate topic guides were used for youth, healthcare workers, and community actors to tailor discussions to specific roles and experiences of each group (S3-S5 Files).

Homogeneity and heterogeneity were carefully balanced within FGDs. Homogeneity was achieved by grouping participants within the same province and similar characteristics (e.g., age, HIV status), while heterogeneity was introduced by including diverse participant demographics and healthcare provider roles. FGDs for youth included participatory activities, with groups of 16 youth further divided into four subgroups of four participants each (S2 File). This design allowed for meaningful engagement and interaction during participatory activities. After data collection, this resulted in a total sample size of 52 youth, stratified by gender. For service providers, FGDs explored their roles, perceptions of youth-friendly healthcare, and the challenges they face in delivering HIV care to youth. Participatory action research (PAR) is a good methodological framework for social inclusion [24]. For that reason, youth and service providers took part in open-ended questions for each theme, as well as participatory activities for decision-making, community mapping, and sorting influential factors for stages 2 and 3 (S1 File).

The data collection process was facilitated by fourteen locally trained data collectors, who were divided into interviewers and note-takers. Measures were taken to ensure participant comfort, including the presence of at least one same-gender team member during sensitive discussions. FGDs were conducted in French and Arabic in a private setting,

lasting between 90 and 120 minutes. All sessions were audio-recorded with participant consent, transcribed verbatim, and imported into ATLAS.ti (version 7.6.3) for analysis [25]. To maintain data quality, a rigorous quality assurance process was implemented. Audio recordings were cross-checked against transcripts by a secondary team member for accuracy, and the first author reviewed a sample of transcripts for consistency. Data collectors underwent comprehensive training to standardize techniques, and iterative feedback loops during debriefing sessions addressed transcription issues.

## Data analysis

Thematic analysis was conducted using both deductive and inductive approaches, chosen for its flexibility and ability to identify, analyze, and report patterns (themes) within the data, providing a rich and detailed account of the participants' experiences and perceptions [26]. Deductive coding was guided by the longitudinal HIV care continuum, a public health model, which provided pre-established codes, including HIV testing, linkage to HIV care and ART, and retention in ART care [27]. These deductive codes were chosen due to their fundamental role in the HIV care continuum and their relevance to the experiences and challenges described by youth in the transcripts [17,27]. During the coding process, these deductive codes served as a foundation, while inductive coding allowed for the identification of additional, emerging themes based on participants' narratives.

The thematic analysis followed the Braun and Clarke's six-phase method to ensure a thorough and rigorous examination of the data [28]. First, familiarization with data involved reading and re-reading the transcripts, while making initial notes and observations. Deductive codes were incorporated at this stage as sensitizing concepts, serving as an analytical framework to organize data. Initial codes were then generated systematically across the dataset, capturing both deductive elements and inductive insights reflecting the data's richness. These codes were subsequently collated into potential themes, grouping related codes to reflect broader patterns of meaning. Themes were reviewed and refined to ensure they accurately represented the data and worked cohesively with the coded extracts and the data as a whole. Each theme was then clearly defined and described, outlining its scope and focus. Finally, a detailed report was produced, illustrating the themes with supporting quotes and their relevance to the research aims. See S1 Table for an overview of the themes and subthemes with sample quotes.

To maintain reliability and cultural accuracy, thematic analysis was conducted in the original languages of the transcripts, French and Arabic, before translating the results into English. This approach preserved linguistic and cultural nuances critical to the findings. Participant verification was achieved by conducting a debriefing session with a subset of participants, who reviewed the preliminary themes and provided feedback to validate the interpretations.

Additionally, two care maps were developed to represent the current pathways to care by HIV status (positive and negative), based on the participatory group activities conducted during youth FGDs. These activities explored decision making processes related to engaging with HIV services [29]. Fig 2A and 2B represent aggregated care maps for HIV negative and positive youth, respectively, while corresponding S2 and S3 Tables describe distinct healthcare resources at each step of care.

## Research trustworthiness

To ensure the trustworthiness of the research, we adhered to the evaluative standards set forth by Lincoln and Guba (1985), which include credibility, transferability, dependability, and confirmability [30]. Credibility was established through prolonged engagement with the data and participant verification. The involvement of locally trained data collectors who were familiar with the cultural context further enhanced the credibility of the data collection process. Transferability was facilitated by providing thick descriptions of the study context, participant demographics, and data collection methods, allowing readers to assess the applicability of the findings to other contexts. Dependability was achieved by maintaining a detailed audit trail of the research process, including the steps taken during data collection, coding, and thematic analysis.

Finally, confirmability was ensured through triangulation, peer debriefing, and maintaining a reflexive journal to document potential biases and how they were managed throughout the research process. Additionally, saturation was achieved at $n=12$ for youth and $n=9$ for service providers, as no new themes or insights emerged from the data. These measures collectively ensured that the findings are trustworthy and accurately represent the experiences of the participants.

## Reflexivity

Reflexivity was a critical component of this study, given the complex sociocultural context of Chad. The research team, comprising individuals from diverse cultural and disciplinary backgrounds, engaged in continuous self-reflection to recognize and mitigate potential biases. As the Principal Investigator and a native of Chad, EB (first author), brought essential local knowledge and cultural sensitivity to the study, particularly regarding the complexity of Chad's healthcare system and socio-economic landscape. Additionally, the collaboration with the RNTAP⁺, composed entirely of Chadian nationals familiar with the local context and HIV, further ensured that the study was deeply rooted in the local reality. The team also included international researchers with extensive expertise in qualitative and HIV research, which guaranteed methodological rigor. The inclusion of locally trained interviewers who were proficient in the local languages and familiar with cultural nuances was essential in capturing authentic data. The entire co-author team was involved in the secondary analysis, ensuring that multiple perspectives enriched the interpretation of the findings, and that cultural and methodological considerations were addressed collaboratively. Regular team meetings were held to discuss and reflect on the data collection process, emerging themes, and any potential biases. The use of a reflexive journal allowed researchers to document their thoughts, decisions, and interactions with participants, which helped in maintaining an awareness of how their perspectives might influence the research. This reflexive practice ensured that the findings were grounded in the participants' realities rather than the researchers' preconceptions.

## Ethics approval

Ethical considerations were paramount in this study, given the sensitive nature of the topic and the vulnerability of the youth participants. Ethical approval was obtained from both the Harvard T.H. Chan School of Public Health's Institutional Review Board (IRB) protocol #IRB23–1743 and the National Committee on Bioethics of Chad (#010/MESRS/SE/SG/2024). Informed consent was obtained from all participants, with additional assent from guardians for participants who were younger than 18 years of age. The consent process involved reading the consent forms aloud and allowing participants to ask questions, ensuring they fully understood their participation rights and the study's purpose. To maintain confidentiality, pseudonyms were used, and all identifying information was removed from the transcripts. Data were securely stored, and only the research team had access to them. Additionally, the research design incorporated measures to minimize potential distress, such as the presence of same-gender interviewers for sensitive topics and providing participants with contact information for counseling services if needed.

Given the stratification of FGDs by HIV status—a decision made to foster comfort and shared understanding among participants—the research team took several additional precautions to prevent stigma and ensure ethical rigor. Youth were informed in advance about the nature of the groupings during the consent process. For youth living with HIV, recruitment was conducted in collaboration with trusted community-based organizations (such as RNTAP+), and participation was entirely voluntary. Care was taken to ensure that participants did not feel singled out, coerced, or exposed during recruitment or discussion. Groups were conducted in neutral, private settings with trained facilitators who emphasized confidentiality and respect at the outset of each session. These measures were designed to strike a balance between generating rich, context-specific data and protecting the dignity and safety of the participants. By embedding these safeguards into the research protocol, the study prioritized participant autonomy, comfort, and confidentiality—especially in light of the heightened stigma associated with HIV in Chad. These practices ensured that the stratification of FGDs did not compromise the ethical integrity of the study and instead created a supportive environment for honest dialogue. This study

followed the COREQ checklist to ensure ethical and methodological rigor in the design, implementation, and reporting of qualitative research. The checklist is included as S6 File.

## Results

### Study participants' characteristics

This study involved a total of 23 FGDs with 52 youth and 48 service providers, including healthcare workers and community actors. Table 1 presents the demographic details of the participants. Among youth, 23 were HIV-positive and 29 were HIV-negative, with an almost equal gender distribution. Most youth were single (92.3%) and had attained at least a high school education. Service providers included 12 doctors, 12 nurses, and 24 community actors, predominantly from public health facilities (79.2%).

### Themes identified

The analysis revealed three overarching themes corresponding to key stages of the HIV care continuum: (1) HIV testing services, (2) linkage to HIV care and ART, and (3) retention on ART. Each theme is supported by subthemes (10 in total) reflecting participants' perspectives and experiences (Table 2).

Youth, whether HIV-negative individuals or those infected with HIV, face particular challenges accessing effective care and achieving successful treatment outcomes. Fig 1 shows HIV care pathways. This was used to guide the organization and discussion of the generation of themes. Fig 1 has 3 different components, including the (1) UNAIDS HIV/AIDS targets for 2030, (2) system successes, and (3) system failures. The targets for 2030 can be read as follows: 95% of people living with HIV know their HIV status, of which (blue arrow), 95% receive treatment, of which (pink arrow), 95% are virally suppressed. To achieve the global 95-95-95 goals, adopted by United Nations Member States in June 2021, it would require system successes [31–33]. When a youth is tested for HIV, two cases can happen: if negative, then this youth is linked to counseling and HIV prevention services, including behavioral and/or biomedical strategies. However, if positive, the youth follows the HIV care continuum. Finally, the third component showcases the system failures due to challenges at different levels, extensively discussed throughout the manuscript.

Fig 2A and 2B depict the care pathways for HIV-negative and HIV-positive youth in Chad, respectively. For HIV-negative youth, the journey begins at the household level and progresses through public health centers, hospitals, and pharmacies, with occasional visits to traditional healers and charlatans. This reflects the blend of traditional and modern healthcare practices in Chad. As youth move through these steps, they may repeatedly visit public health centers and hospitals, highlighting the iterative nature of healthcare-seeking and the need for specialized care at higher-level facilities. This pathway illustrates the complex navigation required within the healthcare system for HIV-negative youth. For HIV-positive youth, the process starts with HIV testing at various institutions, followed by diagnosis and linkage to care through military, public, non-governmental organization, and religious facilities. The retention in care phase involves specialized social clubs and multiple public health institutions.

**Theme 1: HIV testing services. Accessibility of testing:** Youth's access to HIV testing was a critical factor in their engagement with SRH services. Accessibility encompassed the availability of testing facilities, the cost of tests, and the proximity of these centers to their homes or schools. One youth described their experience, saying *"I was tested for HIV five months ago at the BABA Moustapha cultural center. It was part of a workshop. Accessing the service was very simple"* (Participant #11, age range 22–24 years). Despite these examples of easy access, the financial aspect remained a barrier for many, as highlighted by another youth who noted the high costs associated with hospital visits and the need to seek more affordable options like pharmacies. Furthermore, the lack of free additional testing as healthcare often required multiple screenings were not always covered, making it financially burdensome for youth. For instance, a youth shared, *"Going to the hospital for tests costs a lot of money. We young people are looking for shortcuts like going to the*

**Table 1. Sample demographics for youth and service providers.**

| | N | % |
|---|---|---|
| *Youth* | | |
| Total | 52 | 100 |
| Sex | | |
| Male | 28 | 53.8 |
| Female | 24 | 46.2 |
| HIV status | | |
| HIV positive | 23 | 44.2 |
| HIV negative | 29 | 55.8 |
| Religion | | |
| Muslim | 5 | 9.6 |
| Christian | 47 | 93.4 |
| Highest educational level | | |
| No diploma | 7 | 13.5 |
| Enrolled in high school | 28 | 53.8 |
| High school diploma | 11 | 21.2 |
| Bachelor's degree | 6 | 11.5 |
| Marital status | | |
| Single | 48 | 92.3 |
| Married | 4 | 7.7 |
| | **Mean (SD)** | **Range** |
| Age | 19.5 (0.90) | 15-24 |
| *Service providers* | | |
| Total | 48 | 100 |
| Sex | | |
| Male | 29 | 60.4 |
| Female | 19 | 39.6 |
| Provider type | | |
| Nurse | 12 | 25 |
| Physician | 12 | 25 |
| Community actor | 24 | 50 |
| Facility type | | |
| Public | 38 | 79.2 |
| Specialized | 10 | 20.8 |

pharmacy, which is less expensive" (Participant 4, age range 20–22 years). Another mentioned, *"Hospitals are expensive for examinations and due to the lack of resources, we can't do them. If rates can be reduced for us young people, it would be good because we don't have the financial means"* (Participant #1, age range 20–22 years). These challenges were compounded by issues like reagent shortages, which limited the availability of necessary tests, as noted by a community actor: *"On the financial side: access is voluntary; however, we are dealing with reagent shortages, and this has been the case since August 2023 except for pregnant women (prioritized)"* (Community actor, FGD #1).

Youth approached HIV testing influenced by various factors, including family, social environment, and personal health experiences, navigating fragmented care pathways across multiple healthcare sectors (Fig 2). Regular testing for HIV was mentioned as a common practice among HIV-negative youth, often driven by personal or parental initiative, which high-lighted the importance of early detection and periodic testing after risky behaviors such as *"unprotected sex"* (Participant

**Table 2. Themes identified through focus group discussions with participants.**

| Theme | Subtheme | Definition |
|---|---|---|
| **HIV testing** | | |
| | Accessibility of testing | Refers to the ease with which youth can access HIV testing services, including the availability of testing facilities, the cost of testing, and the proximity of these centers to their homes or schools. |
| | Awareness and knowledge | The level of information youth have about HIV testing, including where and how to get tested, the importance of regular testing, and understanding the implications of the test results. |
| | Psychological barriers | The emotional and mental challenges that youth face when considering HIV testing, such as fear of a positive result, stigma associated with being tested, and general anxiety about the testing process. |
| | Support systems | The role of family members, peers, and healthcare providers, including community actors in encouraging and facilitating youth to undergo HIV testing, providing emotional support, and creating a supportive environment. |
| **Linkage to HIV care and ART** | | |
| | Referral processes | The mechanisms and procedures through which youth who test positive for HIV are directed to appropriate HIV care and treatment services. |
| | Initial engagement and barriers to linkage | The initial interactions and obstacles faced by youth when they first attempt to engage with HIV care and treatment services. |
| | Facilitators of linkage | Factors and interventions that help youth successfully connect with HIV care and treatment services after testing positive. |
| **Retention in ART care** | | |
| | Adherence to treatment | The extent to which youth consistently take their antiretroviral medications as prescribed. |
| | Continuous engagement | The ongoing interaction between youth and healthcare providers to ensure they remain in HIV care over the long term. |
| | Retention challenges and support strategies | The difficulties youth face in staying in HIV care and the strategies used to support them in overcoming these challenges. |

#34, age range 18–19 years). Many youths relied on health centers, hospitals, school programs, family referrals, and health campaigns for testing. For instance, one youth shared, *"I am taken by my dad for testing. After testing, we went home, and he came back to get the results"* (Participant #26, age range 18–19 years). Similarly, others described straightforward access to testing through school campaigns and cultural center workshops. Healthcare workers and community actors have played pivotal roles in facilitating access to these services. For example, community actors mediated between young people and healthcare facilities, providing information and support to navigate the testing process: *"My role is to help, accompany, and guide them. In my opinion, it is important to raise awareness in churches, schools, and in the neighborhood"* (Community actor, FGD #2). Besides biomedical strategies, behavioral HIV preventive interventions were also employed since service providers offered advice and counseling to HIV-negative youth. Youth reported receiving advice from *"psycho-social counselors, attending physicians"* (Participant #27, age range 18–19 years), and noted that these providers are key sources of information and support.

**Awareness and knowledge:** The level of awareness and knowledge about HIV testing among youth significantly influenced their likelihood of getting tested. Youth gathered information from various sources, including *"associations working to combat sexually transmitted diseases"* (Participant #9, age range 15–17 years), *"social networks and educational institutions"* (Participant #23, age range 20–22 years). These sources provided crucial information about where and how to get tested, the importance of regular testing, and understanding the implications of the test results. Some youth received referrals from family members, while others took the initiative to seek testing due to health concerns: *"I did not feel well*

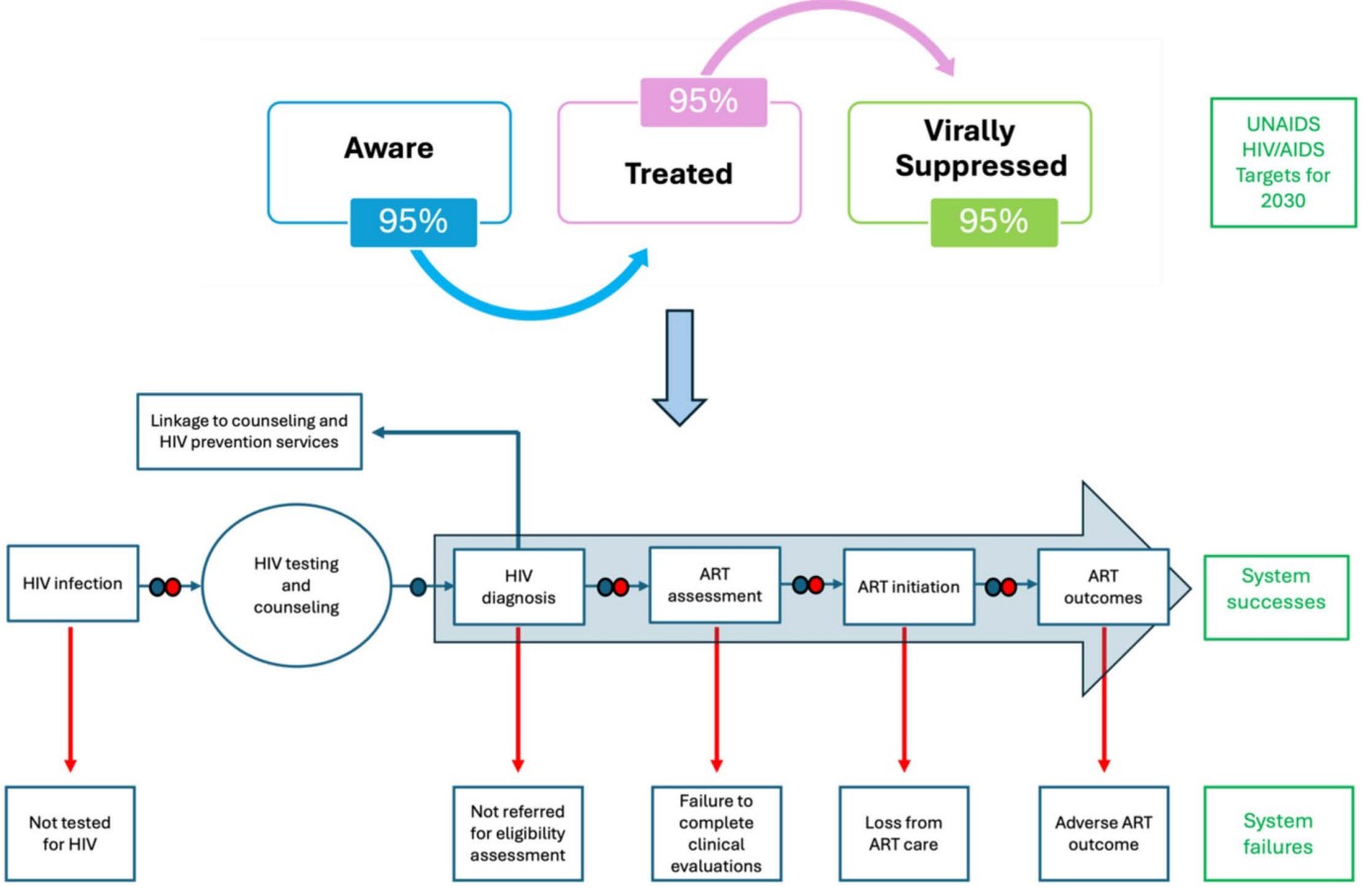

**Fig 1. Overall youth-focused HIV pathway to care in Chad.**

and went straight to the hospital for testing because I did not want to suffer. The fear of dying was great. I took the test *and when I found out it was negative, it prompted me to take preventive measures in the future"* (Participant #4, age range 20–22 years). However, misconceptions and a lack of comprehensive knowledge deterred youth from seeking testing. For instance, one youth mentioned, *"Some people think that once you take the test, you will get infected"* (Participant #2, age range 18–19 years). Another added, *"I thought you could only get tested if you were already sick"* (Participant #1, age range 20–22 years). Healthcare workers were instrumental in educating and advising youth about the importance of regular testing and HIV prevention.

**Psychological barriers:** Psychological barriers played a significant role in youth's decision-making regarding HIV testing. Young women often faced higher barriers to accessing HIV testing due to societal norms and stigma associated with sexual health. Females were more likely to encounter negative reactions from healthcare providers and fear of judgement, which discouraged them from seeking testing. A challenge identified was the mandatory parental presence required for minors due to Chad's legal framework, significantly impacting confidentiality. As one community actor explicitly noted, *"The obstacle is the lack of confidentiality"* and *"family presence is required"* (Community actor, FGD #2). This requirement often heightened anxiety and fear among young people, directly compromising their willingness to seek testing. Additionally, some young women reported being coerced into testing by their parents, sometimes without fully understanding the process or its implications. Healthcare providers attempted to mitigate some of these confidentiality barriers; a healthcare

worker explained their practice: *"When a teenager presents (always accompanied), if the doctor refers them to us for screening, the child is isolated from the parents to find out if sexual intercourse was protected or not"* (Healthcare worker, FGD #1). These practices underscored the tension between legal requirements and the youths' need for privacy and autonomy.

In contrast, young men experienced less societal pressure but still faced significant obstacles. Many young men reported challenges related to the convenience of testing facilities and the fear of receiving their status, which deterred them from seeking testing. A youth shared, *"It's not easy. For us students, we have to be on vacation and every time we take the test, we are afraid of being positive"* (Participant #3, age range 22–24 years). This fear could be overwhelming and prevent youth from getting tested, even when they recognized the importance of knowing their status. Despite the fear and stigma associated with HIV testing, some youth still found the courage to get tested, driven by risky behaviors or symptoms, as expressed by: *"The fear is there, but you have to have the strength to go and take the test"* (Participant #3, age range 22–24 years) or *"I do not think it's easy. You have to be brave. The first day I refused, but once I made love with someone I did not know without protection, I had to find the courage to take the test"* (Participant #2, age range 18-19 years). Addressing these psychological barriers requires creating a supportive and non-judgmental testing environment, providing counseling services to help youth cope with their fears, and normalizing HIV testing as a routine part of healthcare. Healthcare workers and community actors contributed significantly by providing reassurance where youth felt safe and supported during the testing process, as expressed by: *We the psychosocial counselors mediate between young people and the hospital. There are young people who lack information, and some are trying to access these services. So, we are the hyphen, we give them the information"* (Community actor, FGD #1).

**Support systems:** Support systems, including family members, peers, and healthcare providers, were crucial in encouraging youth to undergo HIV testing. The presence of a supportive environment can significantly reduce the fear and stigma associated with testing. For example, one youth mentioned, *"My aunt is a nurse so I ask her to take me to the hospital"* (Participant #1, age range 20–22 years), illustrating the importance of having trusted individuals who can facilitate access to testing (Fig 2). Additionally, schools and community organizations played a vital role in providing information and support, helping to create a culture where HIV testing was seen as a responsible and routine health behavior. For HIV-positive youth, diagnoses often occurred during routine medical visits or emergencies prompted by recurrent illnesses or family-initiated tests (Fig 2). A youth recounted, *"I had pimples on my body and my mom took me to the hospital and after the test, I was diagnosed with HIV"* (Participant #50, age range 15–17 years). Many discovered their status through parental intervention, such as Participant #33 who was screened at age 5 by their mother, or Participant #32, who was screened at age 12 due to frequent illnesses. Youth also encountered diagnoses in unexpected medical contexts, like Participant #30, who tested positive during additional tests following surgery. Table 3 shows selected quotes from HIV-positive youth.

Service providers have been crucial in offering continuous support and follow-up, ensuring that youth remained engaged with the healthcare system. Strengthening these support systems by involving families, peers, and community leaders can enhance youth's willingness to get tested and ensure they receive the necessary support throughout the process. Thus, while these barriers were significant, the support systems and personal determination of youth played crucial roles in their decision to undergo HIV testing.

**Theme 2: Linkage to HIV care and ART.  Referral processes:** This process was shaped by various factors, from initial referral processes to the barriers and facilitators youth encountered. Effective referrals and supportive healthcare professionals were crucial in helping youth transition from testing to treatment. For instance, one youth recounted, *"The psychosocial counselor gave me advice, which helped me to accept treatment"* (Participant #44, age range 20–22 years). Healthcare workers played a vital role in this stage by providing clear guidance and support, ensuring that youth understood the importance of immediate engagement with care, as expressed by, *"It's not the end of the world. You are going to take ARTs and be like other kids. You should come in for the viral load tests in 3-4 months, so if it's*

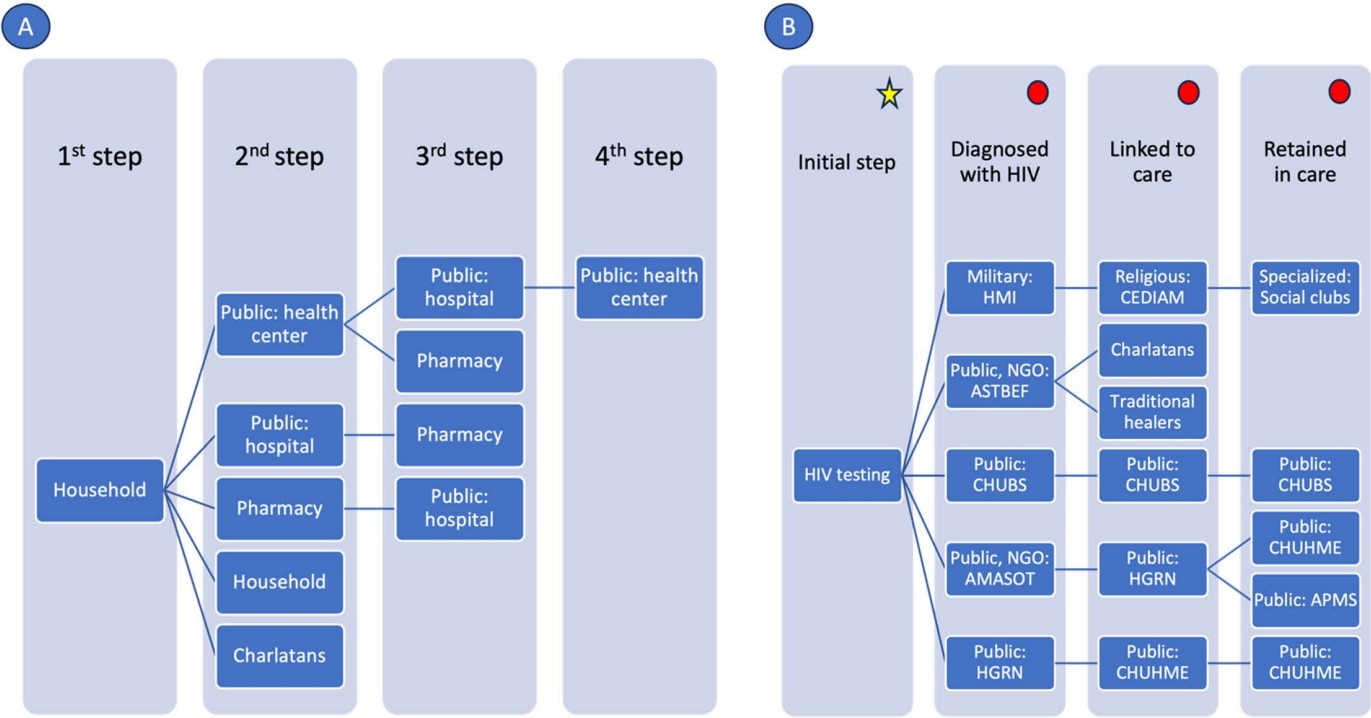

**Fig 2. The current pathways to care experienced by A) HIV-negative and B) HIV-positive youth.**

**Table 3. Quotes from HIV-positive youth on diagnoses following HIV testing with family support.**

| Participants | Age range | Quotes |
|---|---|---|
| Participant 30 | 15-17 years | *For me, it was at the age of 15, following a serious illness that I tested but none of my family had HIV. I was not sexually active; so, it's a bit strange for me to know how I got it.* |
| Participant 32 | 20-22 years | *I was screened at the age of 12 due to recurrent illnesses.* |
| Participant 33 | 18-19 years | *My mother did it when I was 5.* |

*undetectable, you can't transmit the virus to another person anymore, and you are going to do whatever other people do too"* (Healthcare worker, FGD #2).

**Initial engagement:** Youth's first experiences with HIV care were important in determining their long-term engagement with treatment. Positive initial interactions with healthcare providers could significantly influence their decisions to seek care post-diagnosis. One youth highlighted this importance, saying, *"It's difficult at first, but once trust is established between the patient and doctor, everything becomes easy"* (Participant #37, age range 22–24 years). Healthcare workers often provided emotional support and reassurance during this critical period. For example, one healthcare worker mentioned, *"My role is to see the youth living with HIV with a smile. It's very important for me to listen, advise, guide, and lead the person to access his diagnosis"* (Healthcare worker, FGD #3). Additionally, a community health worker said, *"When they learn that they are infected they believe that life is over. But when we tell them that with treatment, they will have*

*HIV-negative children and then, they find hope again"* (Community actor, FGD #1). This supportive approach helped youth overcome initial fears and uncertainties about treatment.

However, despite these efforts, youth faced significant barriers to linkage to HIV care and ART. Financial constraints were a major obstacle, as noted by a youth who said, *"I have no money to pay for medication"* (Participant #44, age range 20–22 years). Due to this situation, some youths were forced to seek non-traditional care options such as charlatans and traditional healers, further endangering their health conditions (Fig 2). Logistical challenges, such as transportation issues and the fear of stigma, also hindered effective linkage. Community actors highlighted the importance of confidentiality and trust, with one noting, *"Dealing with teenagers is difficult and there needs to be an emphasis on confidentiality"* (Community actor, FGD #1). Some youth found it easier to discuss their health with community actors such as peers rather than healthcare workers due to the perceived lack of confidentiality. This was further discussed by a healthcare worker who noted, *"Youth often fear the judgment and lack of confidentiality from healthcare workers"* (Healthcare worker, FDG #1). A community actor explained, *"These youth are not wrong because some youths are not kind and expose children"* (Community actor, FGD #1), which emphasized the need to create a supportive and stigma-free healthcare environment.

**Facilitators of linkage:** Youth's preferences for certain facilities over others, despite geographic proximity, were influenced by several factors. They often favored specialized HIV health facilities and social clubs over public hospitals due to the quality of care, the reception, and the supportive environment provided. As one youth noted, in specialized establishments, *"care is assured and their products are effective because they are subsidized"* whereas in public hospitals, *"care is not total and the welcome is not as good"* (Participant #26, age range 18–19 years). Specialized facilities and social clubs were perceived to offer more comprehensive and empathetic care, free from stigma and discrimination, which was crucial for youth managing sensitive health issues like HIV. This sentiment was echoed by another youth who mentioned that in clubs, *"we feel good"* (Participant #45, age range 20–22 years) whereas in public hospitals, the staff sometimes *"neglect and stigmatize us"* (Participant #44, age range 20–22 years).

Additionally, the availability of trained staff, confidentiality, and effective communication in specialized centers contributed to youth's preferences. Social clubs, in particular, offer a space for open dialogue and guidance, making them attractive to young people seeking both medical and psychosocial support. As one youth described, *"we share everything in words; they encourage us to go to the hospital to get tested and to use the services"* (Participant #4, age range 20–22 years). These facilities also provided a sense of community and emotional support, which public hospitals often lack. Although healthcare workers emphasized the need for continuous education and support, with one stating, *"We must advise them and support them as much as possible"* (Healthcare worker, FGD #2), the specialized facilities and social clubs stood out because they offered tailored environments that prioritized confidentiality, empathy, and comprehensive care. This combination of factors made them particularly appealing to youth who required both medical and psychosocial support in managing their health.

**Theme 3: Retention in ART care. Adherence to treatment:** Youth's commitment to staying on ART was often influenced by their understanding on the importance of consistent medication intake and managing side effects. One youth expressed their determination, stating, *"I agreed to continue ART to avoid transmitting HIV to others and to stay healthy"* (Participant #50, age range 15–17 years). Healthcare workers played a crucial role in this process by providing clear and consistent information about the benefits of adherence and helping manage any side effects that may arise. For example, a healthcare worker noted the importance of therapeutic education, stating, *"It is important to remain clear about the examples of therapeutic education"* (Healthcare worker, FGD #3). This clarity helped youth understand the critical nature of their treatment regimen and reinforced their commitment to maintaining it.

**Continuous engagement:** Retention in ART care also involved regular follow-ups with healthcare providers and continuous engagement with healthcare services. Frequent interactions with healthcare providers helped reinforce the importance of adherence and provide ongoing support. One youth reflected on their routine, mentioning, *"When you feel like you're taking a risk during sexual intercourse, you must be aware and take precautions, including regular follow-ups*

*and adherence to ART"* (Participant #34, age range 18–19 years), illustrating the continuous engagement needed. Healthcare workers emphasized the need to spend adequate time with patients to build trust and ensure they understand their treatment. One healthcare worker stated, *"The patient must always have a little bit of time with the doctor after the first consultation because he is the one who accompanies him through the process"* (Healthcare worker, FGD #4). This ongoing relationship between healthcare providers and youth is essential for maintaining adherence and addressing any issues that may arise during treatment.

**Retention challenges and support strategies:** In previous steps of the HIV care continuum, barriers such as stigma, financial constraints and logistical challenges were identified for HIV testing as well as during the process of linkage to HIV care and ART, respectively. Despite the support from healthcare providers, financial difficulties, transportation issues, and experiences of stigma or discrimination further hindered retention in ART care (Fig 2). For example, logistical issues such as transportation made regular clinic visits challenging. One youth shared, *"It's difficult to access care because of travel"* (Participant #34, age range 18–19 years). Another example involved a teenager who preferred to travel a longer distance to a different health center to avoid stigma, *"There is this teenager who lives in Koundoul (25 km) and prefers to go to the Atrone health center for the pills despite the referral proposal I made to him and this for the simple reason that his sister-in-law works at the Koundoul health center"* (Community actor, FGD #1). Youth have advocated for tailored services to address these challenges, expressing a desire for services that cater specifically to their needs, including tailored hours that accommodate their schedules. For instance, one youth suggested, *"We need special centers for teenagers to make them feel comfortable, with hours that fit our school and work schedules"* (Participant #8, age range 15–17 years). This sentiment was echoed by community actors who recognized the need for flexible service hours: *"It would be important to create special structures with everything in it, including tailored hours that allow youth to access care without disrupting their daily routines"* (Community actor, FGD #1).

Youth often stopped treatment when they started feeling better, which disrupted their long-term care. Healthcare workers have responded to these needs by providing longer drug stocks to youth who have earned their trust, which helps to maintain continuity in their treatment. One community actor explained, *"We call them (by phone) if they don't keep the appointments. We're going to pick them up (move). At first, they are put on ARVs and when they get better, they leave and only come back when there is a complication"* (Community actor, FGD #1). Another one shared, *"When we gain their trust and they become more attached to us than their own parents, we can trust them with 3 to 6 months of drug stocks"* (Community actor, FGD #1). This approach not only addressed the logistical challenges but also helped ensure that youth remain engaged in their ART regimen.

## Discussion

Findings from this study provide insights into the barriers and facilitators experienced by youth in Chad across the HIV care continuum, based on participants' perceptions and lived experiences. These findings underscore the importance of understanding youth-specific challenges to inform strategies for improving access to and retention in HIV care services. The pathway to SRH and HIV care for youth should ideally be youth-centered, incorporating accessible and youth-friendly services, supportive environments, and targeted interventions addressing social determinants of health [34–37]. This aligns with broader recommendations in global health literature advocating for inclusive, context-sensitive approaches to SRH and HIV care for youth [36–39]. However, the participants' accounts reveal a disconnect between this ideal and the reality faced by many youths in Chad, where financial constraints, logistical challenges, and stigma present significant obstacles to care. Our findings align with recent literature emphasizing adaptive sensemaking as a critical process through which youth navigate fragmented health systems, especially in resource-limited and high-stigma environments [40]. Integrating the adaptive sensemaking into HIV service design—by creating enabling environments where youth can process, co-construct meaning, and engage with care on their own terms—may further improve service uptake and continuity [40].

Accessibility remains a fundamental barrier to HIV testing and care for youth. Participants revealed gendered experiences that shaped their access to HIV care. Young women highlighted societal norms and stigma as critical barriers to seeking HIV testing, reflecting deeply entrenched gender inequities. Young men, on the other hand, reported challenges such as the lack of convenient of testing facilities and fear of receiving a positive diagnosis, which discouraged engagement with HIV testing services. These gendered narratives emphasize the need for targeted interventions that address the unique barriers faced by both young men and women. Stigma emerged as a pervasive barrier to care and treatment adherence among youth living with HIV, as reflected in participants' experiences. The fear of being stigmatized often prevented youth from accessing services or disclosing their status, which could compromise adherence to ART. This finding is consistent with prior research conducted in sub-Saharan Africa, which identifies stigma as a critical challenge in achieving optimal HIV care outcomes [4,41–43]. Youth participants further emphasized the importance of integrating testing services into familiar and convenient locations, in part to mitigate stigma. Integrating HIV testing into a multi-disease approach, as suggested by youth has been shown to increase efficiency while reducing stigma in SSA, especially among men [44,45]. Youth frequently described financial and logistical barriers as key hindrances to accessing HIV testing and care. For example, many youths cited the cost of transportation and healthcare services as prohibitive. Others described logistical challenges, such as the limited availability of testing sites or the irregular supply of medical reagents, which disrupted their access to care. To address these issues, youth and service providers suggested the establishment of more youth-specific centers in close proximity. These centers should be strategically located to minimize travel distances and associated costs, making it easier for youth to access the services they need. By reducing financial and logistical barriers and providing services within their communities, such as youth-focused centers can play a pivotal role in improving HIV care engagement and outcomes for this demographic.

The responsiveness of care is crucial for maintaining youth's engagement in HIV services. Youth in our study reported that negative interactions with healthcare providers, long waiting times, and lack of confidentiality deterred them from seeking and continuing care. Similar challenges have been reported by studies, which highlight how health-system and structural barriers further complicate youth engagement in HIV care [46]. However, solutions exist that can mitigate these challenges. Building on the creation of specific centers for youth, these spaces need to be safe and supportive environments, delivering screening services, psychosocial support, and adhering to youth-specific treatment protocols. Additionally, youth highlighted the need for healthcare providers to be more understanding and respectful, which can be facilitated through targeted training programs for healthcare workers on youth-friendly practices. Research has shown that engaging youth in the design of HIV care interventions significantly enhances their engagement and acceptance of services. Furthermore, Akama et al. (2023) demonstrated that youth-led initiatives can improve care engagement by making services more responsive to youth's needs [47].

Ensuring clinical effectiveness involves providing comprehensive and continuous care tailored to the specific needs of youth. Continuous, and specialized training for healthcare workers was identified as essential for equipping providers to deliver youth-specific services. Training should include components on communication, confidentiality, and youth health dynamics to foster empathetic, respectful, and effective care. Recent research has demonstrated the effectiveness of such approaches in improving linkage to and retention in care [48,49]. Furthermore, in youth-centered design, the care provided is based on the best available evidence and is specifically tailored to the unique developmental and psychosocial aspects of youth health. This involves using interventions that have been proven effective for this age group and incorporate feedback from youth to adapt clinical practices and treatment protocols. Continuity of care – ensuring that youth do not fall out of the system – is enhanced by such feedback loops and youth involvement. For instance, studies have shown that youth-friendly service models, coupled with human-centered design approaches, enhance retention in care among youth [50,51].

Finally, tailored service delivery requires a holistic approach that integrates biomedical, behavioral, psychosocial, and structural perspectives. Youth in our study emphasized the positive impact of having peer educators and psycho-social

counselors who provided both information and emotional support, making the healthcare experience more relatable and less intimidating. Recent research supports this integrated approach, showing that peer-led interventions and the inclusion of psycho-social support within youth-friendly spaces significantly improve access and engagement, as well as enhance linkage to care and retention rates among youth [50–54]. Service providers similarly advocated for community-based programs that engage families and community leaders to reduce stigma and support adherence to treatment. For example, interventions like Community Adolescent Treatment Supporters (CATS) in Zimbabwe have been shown to improve adherence, retention, and psychosocial well-being among HIV-positive youth [55]. Further studies have shown that these community-based and multi-faceted interventions are essential for addressing the complex needs of youth [56]. Such approaches promote a supportive and effective environment that fosters better health outcomes and improves engagement in HIV care.

In addition to service delivery improvements, the national policy environment in Chad plays an important role in shaping youth access to HIV care. Notably, Chad currently does not have a specific law allowing minors to independently consent to HIV services – in practice, adolescents under 18 must obtain parental or guardian consent for HIV testing and treatment [57]. Such consent requirements are common in many countries and are recognized as barriers to youth HIV services [58]. Indeed, the need for parental involvement was evident in our study, where participants noted that "family presence is required" for teenagers seeking testing. These policy-related barriers can discourage youth from seeking timely testing and care due to confidentiality concerns. On the other hand, the Chadian government has undertaken supportive measures, such as providing free HIV care for people living with HIV and adopting various policy plans focused on HIV, including pediatric and adolescent HIV care [59]. These initiatives, backed by high-level political commitment demonstrate a willingness to improve access to HIV services for young people [60]. Aligning national policies with youth-friendly approaches – for example, revising consent policies to empower mature adolescents to seek HIV testing and care independently – could significantly improve uptake of services. Additionally, fully implementing youth-centered strategies and ensuring they are resourced will help translate high-level commitments into real improvements on the ground. In summary, fostering a supportive policy and legal framework, one that protects youth's rights to health, confidentiality, and informed consent, is essential alongside programmatic interventions, so that youth in Chad can more easily access the HIV services they need without unnecessary barriers.

Despite the valuable insights gained from this study, several limitations should be acknowledged. First, while the study involved 23 FGDs with a diverse sample, participants were recruited from specific urban contexts, which may limit the transferability of findings to rural settings or other provinces. Expanding future research to include rural areas would provide a broader understanding of youth experiences across diverse contexts. Second, reliance on self-reported data introduces potential biases, as participants may have provided socially desirable responses or struggled to recall their experiences. Triangulating self-reported data with observations or other data sources could enhance the validity of findings. Third, while thematic analysis was conducted in the original languages (French and Arabic) to preserve linguistic and cultural nuances, translation into English might have introduced subtle shifts in meaning. Efforts were made to mitigate this through a rigorous quality assurance process. Finally, the study's cross-sectional design captures a snapshot of participants' experiences at a single point in time. Future qualitative studies with longitudinal designs could provide deeper insights into how experiences and challenges evolve over time, particularly in relation to long-term ART adherence and engagement in care.

## Conclusion

This study explored the experiences and perspectives of youth regarding their pathways to care, focusing on their journey from HIV diagnosis to ART initiation, retention, and adherence. Findings revealed that youth face a range of barriers, including stigma, financial constraints, logistical challenges, and negative interactions with healthcare providers, which hinder their access to and retention in HIV care. Facilitators such as supportive healthcare environments, peer educators,

psychosocial support, and specialized youth-friendly services were identified as critical to improving youth engagement in care. These findings emphasize the need to address both structural and interpersonal barriers to enhance the HIV care continuum for youth in Chad. Based on these findings, it is recommended that healthcare systems prioritize the development of youth-centered care models that integrate psychosocial support, address stigma, and provide youth-friendly training for healthcare providers. Community-based programs involving families and community leaders can further reduce stigma and enhance support systems. By integrating these strategies, healthcare systems can create an enabling environment that addresses the unique challenges faced by youth, ensuring better access to and retention in HIV care. Future research should build on these insights to evaluate the long-term impact of youth-friendly interventions in diverse contexts.

## Supporting information

**S1 File. Stages and processes of the constructivist grounded approach employed for parent study to enable a youth sensemaking outcome on youth-centered decision making in HIV care in Chad.**
(DOCX)

**S2 File. Flowchart of sub-divided participants in each group.**
(DOCX)

**S3 File. Focus group discussion topic guide for youth.**
(DOCX)

**S4 File. Focus group discussion topic guide for healthcare workers.**
(DOCX)

**S5 File. Focus group discussion topic guide for community actors.**
(DOCX)

**S6 File. COREQ checklist.**
(PDF)

**S1 Table. Themes identified through focus group discussions with participants.**
(DOCX)

**S2 Table. Healthcare sectors identified for the current pathways to care for HIV-negative youth.**
(DOCX)

**S3 Table. Healthcare sectors identified for the current pathways to care for HIV-positive youth.**
(DOCX)

## Acknowledgments

We thank the study participations for their willingness to share their experiences and perspectives, which has provided an essential foundation for our research. We are also grateful to the Fostering Diversity in HIV Research Program (R25MH119857).

## Author contributions

**Conceptualization:** Esias Bedingar, Ngarossorang Bedingar, Rifat Atun, Aisha K. Yousafzai.

**Data curation:** Esias Bedingar, Ferdinan Paningar, Ngarossorang Bedingar, Eric Mbaidoum, Naortangar Ngaradoum.

**Formal analysis:** Esias Bedingar, Ferdinan Paningar.

**Funding acquisition:** Esias Bedingar, Ngarossorang Bedingar, Rifat Atun, Aisha K. Yousafzai.

**Investigation:** Esias Bedingar, Ngarossorang Bedingar, Eric Mbaidoum, Naortangar Ngaradoum.

**Methodology:** Esias Bedingar, Ngarossorang Bedingar, Eric Mbaidoum, Naortangar Ngaradoum, Rifat Atun, Aisha K. Yousafzai.

**Project administration:** Esias Bedingar, Ferdinan Paningar, Ngarossorang Bedingar, Eric Mbaidoum, Naortangar Ngaradoum.

**Resources:** Esias Bedingar.

**Supervision:** Esias Bedingar, Ferdinan Paningar, Ngarossorang Bedingar.

**Validation:** Esias Bedingar.

**Visualization:** Esias Bedingar.

**Writing – original draft:** Esias Bedingar.

**Writing – review & editing:** Esias Bedingar, Rifat Atun, Aisha K. Yousafzai.

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
