## [Decision Letter · Decision Letter 0]

Dear Dr. Bedingar,

Thank you for submitting your manuscript to PLOS ONE. After careful consideration, we feel that it has merit but does not fully meet PLOS ONE’s publication criteria as it currently stands. Therefore, we invite you to submit a revised version of the manuscript that addresses the points raised during the review process.

Upon review, I am in general agreement with the first reviewer that there is additional work required. Given the time it has taken to secure reviewers and the need for a second round of review, I encourage you to take time to reflect on reviewer 1's feedback, which is detailed, and submit an updated version. Should you disagree with any points in the review, ensure that you include a clear response. In addition to the reviewer comments, I suggest the following revisions: 1. Review and update the labelling of your supplementary files within the text, particularly S1.2. While your section on reflexivity is great, please provide additional clarify on who was involved in the secondary analysis. Was this the entire co-author team or a subset?3. Remove periods after the headings for tables and figures4. According to the journal requirements, your Tables should placed within the body of the manuscript directly after the paragraph in which it is first cited (read order) instead of Supplemental Files. Tables require a label (e.g., “Table 1”) and brief descriptive title to be placed above the table in the manuscript.==============================

We look forward to receiving your revised manuscript.

Kind regards,

Sara Jewett Nieuwoudt, Ph.D, MPH

Academic Editor

PLOS ONE

Journal Requirements: 

The study received funding from the Tessa Jowell Fellowship for Doctoral Research and Chad’s Prime Minister’s Office.  

We are grateful for the collaboration and support received from the Blue Cross Chad, Rose Service Learning Fellowship, Fostering Diversity in HIV Research Program, Tessa Jowell Fellowship for Doctoral Research, Gamble Scholarship, and Chad’s Prime Minister’s Office. We thank the study participations for their willingness to share their experiences and perspectives, which has provided an essential foundation for our research. The study received funding from the Tessa Jowell Fellowship for Doctoral Research and Chad’s Prime Minister’s Office. 

The study received funding from the Tessa Jowell Fellowship for Doctoral Research and Chad’s Prime Minister’s Office.

6. See earlier Editor Comments on how to align with the journal requirements.

Reviewers' comments:

Reviewer's Responses to Questions

**Comments to the Author**

1. Is the manuscript technically sound, and do the data support the conclusions?

Reviewer #1: Partly

2. Has the statistical analysis been performed appropriately and rigorously?

Reviewer #1: N/A

3. Have the authors made all data underlying the findings in their manuscript fully available?

Reviewer #1: Yes

4. Is the manuscript presented in an intelligible fashion and written in standard English?

Reviewer #1: No

Reviewer #1: Overall, the manuscript has a potential, and with adjustments, it could make a valuable contribution to the field. I recommend that the authors should consider the following suggestions

The primary issue is the predominant use of quantitative language to describe what is fundamentally qualitative research. I recommend that the authors revise how they present their findings using qualitative descriptors and narratives that better capture the depth and richness of their data.

One gets the impression that the primary study generated good qualitative data. The proposed secondary analysis should be clearly described in terms of the research question that informed their analysis and the details of the research methods.

The authors should align their language and presentation style with the qualitative nature of their research to maintain coherence throughout the manuscript.

**Do you want your identity to be public for this peer review?** For information about this choice, including consent withdrawal, please see our Privacy Policy

Reviewer #1: No

---

## [Author Response · Author response to Decision Letter 1]

30 Dec 2024

Dr. Sara Jewett Nieuwoudt

Academic Editor

PLOS ONE

December 24th, 2024

Dear Dr. Sara Jewett Nieuwoudt,

Thank you for your insightful and constructive comments regarding our manuscript. We have taken a great consideration for each of the comments provided. We provide a detailed response to each of the points raised:

1. Editor’s comments

a. Reviewed and updated the labelling of my supplementary files within the text.

b. Provided additional clarification on who was involved in the secondary data analysis.

c. Removed periods after the headings for tables and figures.

d. Placed Tables within the body of the manuscript directly after the paragraph in which it is first cited.

2. Journal requirements

a. Ensured that the manuscript meets PLOS ONE’s style requirements, including those for file naming.

b. Created a new section called “Funding” to state the financial disclosure, and the role the funders took in the study.

c. Amended the Role of Funder statement within my cover letter.

d. Updated the Data Availability statement.

e. Included captions for my Supporting Information files at the end of my manuscript and updated in-text citations to match accordingly.

3. Reviewer’s comments

a. Abstract

i. Clarified the study design to specify that the study is a secondary data analysis based on a parent study that employed a Grounded Theory design to develop theory inductively.

ii. Removed the term “significant” to avoid exaggeration as it cannot be supported by qualitative data.

iii. Added findings from healthcare workers (service providers), emphasizing their perspectives on barriers, facilitators, and their roles in the HIV care continuum.

iv. Rephrased the statement to make it a clear recommendation rather than a prescriptive assertion.

b. Introduction

i. Rephrased the study aim to reflect the qualitative nature of the research.

c. Sampling

i. Defined “diverse” and “relevant” in the context of this study.

ii. Addressed the random selection question for the selection of participants.

iii. Revised the paragraph to mention three categories instead of focus group discussions.

iv. Re-wrote all numbers from one to ten from numbers to letters.

d. Data collection

i. Moved all details about the recruitment criteria, planned participant numbers, and random selection process to Sampling.

ii. Clarified definitions of diversity and relevance in Sampling.

iii. Organized details about service provider categories and facility eligibility under Sampling.

iv. Focused on the process and logistics of conducting FGDs.

v. Highlighted participatory activities and their role in data enrichment.

vi. Kept descriptions of audio recording, transcription, and quality assurance in this section.

e. Data analysis

i. Clarified how deductive codes were used as a framework in the initial stages of coding and complemented by inductive coding to capture emergent themes.

ii. Explicitly stated the research aims for the secondary data analysis.

iii. Added a description of how participant feedback was sought and used to validate findings.

iv. Replaced the word “guaranteed” with a more accurate description of efforts to ensure reliability and cultural accuracy.

v. Retained the explanation of working with original languages while empathizing the rationale behind this approach.

f. Results

i. Added an introductory paragraph to contextualize the findings.

ii. Described demographics and referenced clearly.

iii. Expanded on participant demographics and their relevance to study findings.

iv. Created descriptive headings reflecting the qualitative nature of the findings, removing numerical labeling for subthemes.

v. Ensured findings were described before referencing tables and figures, per reviewer feedback.

g. Discussion

i. Rephrased statements to reflect that findings are based on participants’ experiences and perceptions, avoiding generalizations not supported by qualitative data.

ii. Rephrased statements that implied comparison, measurement, or quantitative assessments.

h. Conclusion

i. Rephrased conclusion to link findings to the aim.

ii. Avoided opinion-like statements, grounding conclusions in the data.

iii. Maintained a focus on actionable recommendations while emphasizing the study’s contribution to knowledge.

We are confident that these revisions have strengthened our manuscript and addressed the concerns raised. We hope that the revised manuscript will now be considered favorably for publication at PLOS ONE.

Thank you once again for your constructive feedback. We are available to address any queries or provide additional information as needed. Please address all correspondence to: esias_bedingar@g.harvard.edu or esias.bedingar@gmail.com.

Sincerely,

Esias Bedingar, SM, PhD on behalf of the author team.

---

## [Decision Letter · Decision Letter 1]

Dear Dr. Bedingar,

Thank you for submitting your manuscript to PLOS ONE. After careful consideration, we feel that it has merit but does not fully meet PLOS ONE’s publication criteria as it currently stands. Therefore, we invite you to submit a revised version of the manuscript that addresses the points raised during the review process.

**ACADEMIC EDITOR:**

We look forward to receiving your revised manuscript.

Kind regards,

Sara Jewett Nieuwoudt, Ph.D, MPH

Academic Editor

PLOS ONE

Journal Requirements:

Reviewers' comments:

Reviewer's Responses to Questions

**Comments to the Author**

Reviewer #2: (No Response)

2. Is the manuscript technically sound, and do the data support the conclusions?

Reviewer #2: Yes

3. Has the statistical analysis been performed appropriately and rigorously?

Reviewer #2: N/A

4. Have the authors made all data underlying the findings in their manuscript fully available?

Reviewer #2: No

5. Is the manuscript presented in an intelligible fashion and written in standard English?

Reviewer #2: Yes

Reviewer #2: In this paper, the authors analyze secondary qualitative data to understand the barriers and facilitators to HIV care for adolescents.

The paper is well-written and addresses an important public health issue. Below I highlight a few concerns about the framing of the paper that I hope the authors find helpful

1) To help the reader understand the context in Chad, it would be helpful to speak about policies around young people’s access to services, including any age of consent restrictions when it comes to getting HIV testing and other HIV care services

2) While a qualitative study is suitable given the study aim, one aspect that is not sufficiently discussed under the ethical implications section is the suitability of FGDs for the interactions with adolescents particularly as the groups were divided by HIV status. Given the potential for stigma, it would be important for the authors to what measures were taken to ensure that young people were comfortable being put in groups based on HIV status. Were they aware of this grouping?

3) Given that care for HIV negative adolescents is centered around HIV testing, it feels like the paper would be more focused if it was on pathways to care for youth with HIV.

4) The discussion is sparse on references with several statements needing relevant citations (e.g., line 495 - Similar challenges have been reported by studies, highlighting these health-system and structural barriers, further complicating youth’s engagement in HIV care.)

Minor suggestion

Since the paper mostly uses the terms “youth” and “young people” rather than “adolescents”, when speaking to responsive care, they should use the term “youth-friendly” rather than “adolescent-friendly.” Similarly, they should use one term consistently (e.g., lines 225-227, 511-515 the authors switch to using the term “adolescent”)

There are acronyms listed that are not used in the text e.g., LGBTQ+

**Do you want your identity to be public for this peer review?** For information about this choice, including consent withdrawal, please see our Privacy Policy

Reviewer #2: No

---

## [Author Response · Author response to Decision Letter 2]

5 May 2025

Dear Dr. Sara Jewett Nieuwoudt,

Thank you again for your constructive and thoughtful comments regarding our manuscript. We have carefully reviewed and addressed all feedback from both the editorial team and the reviewer. Below, we provide a point-by-point summary of the changes made to the manuscript, including a response to the most recent request regarding references accuracy and completeness.

1. Editor’s comments

a. Made a final round of revisions based on the feedback of the second reviewer.

2. Journal requirements

a. Added additional references and edited the reference list. The references are #34 (lines 824-826), #40 (lines 843-845), #51 (lines 882-884), #52 (lines 885-887), #53 (lines 888-890), and #54 (lines 891-893).

3. Reviewer #2 comments

a. Introduction

i. Added a section on the legal and policy-related obstacles to accessing HIV services.

b. Methods

i. Addressed comment #2 by adding more information on the stratification of FGDs by HIV status in the Ethical approval section.

c. Discussion

i. Addressed comment #4 by reviewing and updating our reference list to ensure it is complete and accurate.

d. Minor suggestion

i. Changed the term “adolescents” to “youth” and “young people” when speaking to responsive care.

ii. Removed the acronym “LGBTQ+” from the acronym list.

We are confident that these revisions have substantially strengthened our manuscript and fully addressed the reviewer and editorial feedback. We respectfully resubmit our revised manuscript and hope it will be considered favorably for publication in PLOS ONE. Please do not hesitate to contact us with any further questions.

Warm regards,

Esias Bedingar, SM, PhD on behalf of the author team.

---

## [Decision Letter · Decision Letter 2]

Dear Dr. Bedingar,

Thank you for submitting your manuscript to PLOS ONE. After careful consideration, we feel that it has merit but does not fully meet PLOS ONE’s publication criteria as it currently stands. Therefore, we invite you to submit a revised version of the manuscript that addresses the points raised during the review process.

We look forward to receiving your revised manuscript.

Kind regards,

Sara Jewett Nieuwoudt, Ph.D, MPH

Academic Editor

PLOS ONE

Journal Requirements:

Reviewers' comments:

Reviewer's Responses to Questions

**Comments to the Author**

Reviewer #2: (No Response)

2. Is the manuscript technically sound, and do the data support the conclusions?

Reviewer #2: Yes

3. Has the statistical analysis been performed appropriately and rigorously?

Reviewer #2: N/A

4. Have the authors made all data underlying the findings in their manuscript fully available?

Reviewer #2: No

5. Is the manuscript presented in an intelligible fashion and written in standard English?

Reviewer #2: Yes

Reviewer #2: Thank you to the authors for their thoughtful revisions to the manuscript. The paper is well-written and offers valuable insights for HIV programs focused on young people. I have two minor suggestions:

I appreciate the enhanced discussion of the implications of the legal age of consent in both the introduction and discussion sections. While the discussion notes that parental presence during testing was identified as a challenge (page 28), this finding is not explicitly presented in the results. Please add a sentence or two detailing these findings in the results section.

Regarding the acronym list: as previously mentioned, it should only include acronyms used within the text. Acronyms such as FCAS, CDC, and CGT are not used and should be removed. Additionally, "NGO" appears only once (page 15) and is not defined. I recommend spelling out "non-governmental organization" on page 15 and removing "NGO" from both the text and the acronym list.

**Do you want your identity to be public for this peer review?** For information about this choice, including consent withdrawal, please see our Privacy Policy

Reviewer #2: No

---

## [Author Response · Author response to Decision Letter 3]

1 Jun 2025

Dear Dr. Sara Jewett Nieuwoudt,

Thank you again for your constructive and thoughtful comments regarding our manuscript. We have carefully reviewed and addressed all feedback from both the editorial team and the reviewer. Below, we provide a point-by-point summary of the changes made to the manuscript, including a response to the most recent request regarding references accuracy and completeness.

1. Editor’s comments

a. Made a final round of revisions based on the feedback of the second reviewer.

b. There is no explicit use of the word “adolescent” in the abstract, and the abstract already aligns with the youth-friendly framing, consistently using the term “youth”.

2. Reviewer #2 comments

a. Minor suggestion

i. Although it was already explicit through the quotes used on page 18, I edited the paragraph to make it more explicit.

We are confident that these revisions have substantially strengthened our manuscript and fully addressed the reviewer and editorial feedback. We respectfully resubmit our revised manuscript and hope it will be considered favorably for publication in PLOS ONE. Please do not hesitate to contact us with any further questions.

Warm regards,

Esias Bedingar, SM, PhD on behalf of the author team.

---

## [Editor Report · Decision Letter 3]

Rethinking HIV care for youth: Insights from qualitative research with youth in Chad.

PONE-D-24-34018R3

Dear Dr. Bedingar,

We’re pleased to inform you that your manuscript has been judged scientifically suitable for publication and will be formally accepted for publication once it meets all outstanding technical requirements.

Kind regards,

Sara Jewett Nieuwoudt, Ph.D, MPH

Academic Editor

PLOS ONE

Additional Editor Comments (optional):

Congratulations! This is an important contribution to the literature, particularly Chad. Thank you for your patience and commitment to publishing with us. While I've noted that Figure 1 appears twice at the end, but this will be handled through the publishing process. I look forward to seeing this in print.

---

## [Editor Report · Acceptance letter]

PONE-D-24-34018R3

PLOS ONE

Dear Dr. Bedingar,

I'm pleased to inform you that your manuscript has been deemed suitable for publication in PLOS ONE. Congratulations! Your manuscript is now being handed over to our production team.

Kind regards,

on behalf of

Dr. Sara Jewett Nieuwoudt

Academic Editor

PLOS ONE